# A gene expression map of shoot domains reveals regulatory mechanisms

Caihuan Tian [1], Ying Wang[2], Haopeng Yu [1,2,3], Jun He[1,6], Jin Wang [1,2,7], Bihai Shi[1,2,8], Qingwei Du[1,2], Nicholas J. Provart [4], Elliot M. Meyerowitz[5] & Yuling Jiao [1,2]

Gene regulatory networks control development via domain-specific gene expression. In seed plants, self-renewing stem cells located in the shoot apical meristem (SAM) produce leaves from the SAM peripheral zone. After initiation, leaves develop polarity patterns to form a planar shape. Here we compare translating RNAs among SAM and leaf domains. Using translating ribosome affinity purification and RNA sequencing to quantify gene expression in target domains, we generate a domain-specific translatome map covering representative vegetative stage SAM and leaf domains. We discuss the predicted cellular functions of these domains and provide evidence that dome seemingly unrelated domains, utilize common regulatory modules. Experimental follow up shows that the RABBIT EARS and HANABA TARANU transcription factors have roles in axillary meristem initiation. This dataset provides a community resource for further study of shoot development and response to internal and environmental signals.

[1] State Key Laboratory of Plant Genomics, Institute of Genetics and Developmental Biology, Chinese Academy of Sciences, Beijing 100101, China. [2] College of Life Sciences, University of Chinese Academy of Sciences, Beijing 100049, China. [3] West China Biomedical Big Data Center, West China Hospital/West China School of Medicine, and Medical Big Data Center, Sichuan University, Chengdu 610041, China. [4] Department of Cell and Systems Biology, Centre for the Analysis of Genome Evolution and Function, University of Toronto, Toronto, ON M5S 3B2, Canada. [5] Howard Hughes Medical Institute and Division of Biology and Biological Engineering, California Institute of Technology, Pasadena, CA 91125, USA. [6] Present address: Horticultural Plant Biology and Metabolomics Center, Institute of Science and Technology, Fujian Agricultural and Forestry University, Fuzhou, Fujian 35002, China. [7] Present address: Zhongshan School of Medicine, Sun Yat-sen University, Guangzhou, Guangdong 510080, China. [8] Present address: Laboratoire Reproduction et Developpement des Plantes, Univ Lyon, ENS de Lyon, UCB Lyon 1, CNRS, INRA, 69342 Lyon, France. Correspondence and requests for materials should be addressed to Y.J. (email: yljiao@genetics.ac.cn)

In multicellular eukaryotes, including plants, a majority of genes show differential expression in various tissues and domains. The development and function of plant tissues rely on constant interactions among distinct and nonequivalent domains. The formation of these domains from their ancestors relies on the reorganization of their gene regulatory networks. To understand how cells work and how they interface with the environment, it is useful to acquire quantitative information on transcriptomes and translatomes (translating messenger RNAs (mRNAs)) at cellular and cell-type resolution. Technologies to achieve this have been developed and have substantially expanded our understanding of cell identity and function in plants[1–6].

The shoot apical meristem (SAM) consists of a central zone (CZ) of pluripotent stem cells in the center, an organizing center (OC) beneath the CZ, and a peripheral zone (PZ) surrounding the CZ. The stem cell niche is maintained by the OC which provides stem cell-promoting cues[7–9]. New organs such as leaves and flowers are generated from the periphery.

Leaf primordia initiate from the PZ of the SAM during vegetative growth. Adaxial–abaxial (dorsoventral) asymmetries result in a flattened structure with adaxial–abaxial differences in tissue arrangement and sufficient area for photosynthetic light harvesting and gas exchange. Previous work has defined transcriptional regulatory networks of abaxial- and adaxial-promoting protein and microRNA encoding genes[10–12]. Genes expressed in the adaxial domain can be suppressed by those expressed in the abaxial domain, and vice versa. Auxin also contributes to leaf patterning, and translates adaxial–abaxial polarity into leaf blade expansion[13]. Axillary meristems (AMs) initiate in the leaf axils between the SAM and developing leaves. A transcriptional regulatory network and hormone responses are involved in AM initiation[14–16]. AMs have the same developmental potential as the SAM, making the whole plant a ramifying system.

RNA profiling of mutant and over-expression seedlings with leaf polarity defects has been used to identify regulators of leaf development[17–19], but these are profiles of heterogeneous tissues. Cell type-specific profiling approaches have been used to understand the floral meristem (FM) that develops into flowers[3,19–21]. However, the FM has a determinate cell fate, making it different from an indeterminate SAM.

In this study, we provide a domain-specific gene expression map covering key SAM and leaf domains, allowing direct comparison among shoot domains. These domain resolution expression profiles allow us to identify dominant signatures associated with each domain, systems-level principles of gene regulation, and potential regulators of cell functions.

## Results

**Labeling and profiling SAM and leaf domains using TRAP-seq.** We used the *CLAVATA3* (*CLV3*), *UNUSUAL FLORAL ORGANS* (*UFO*), and *WUSCHEL* (*WUS*) promoters to label the CZ, PZ, and OC of the SAM, respectively[22–24]. For leaf domains, we used the *ASYMMETRIC LEAVES2* (*AS2*) and *FILAMENTOUS FLOWER* (*FIL*) promoters to label adaxial and abaxial cells[25,26]. The *AS2* and the *FIL* expression domains encompass young and older leaves, making their temporal expression different from the *ASYMMETRIC LEAVES1* (*AS1*) promoter, which is mostly active in young leaf primordia earlier than $P_6$[25], and was previously used to label the entire leaf[16]. In addition, we used the *Arabidopsis thaliana MERISTEM LAYER1* (*ATML1*) promoter to label epidermal cells[27] and the *PETAL LOSS* (*PTL*) promoter to label leaf margin cells[28] (Fig. 1a).

We used most of these promoters to drive the expression of a fusion of the large subunit ribosomal protein L18 with N-terminal His and FLAG epitope tags (*HF-RPL18*) through the LhG4/pOp

transactivation system. For the *CLV3* promoter, we used the promoter to directly drive *HF-RPL18* expression. To confirm domain specificity, we used pOp-driven GFP or GUS in transactivation lines, or an immunohistochemical assay to detect HF-RPL18 protein. As shown in Supplementary Fig. 1, all of the promoters have faithful expression compared with the endogenous genes, except for *UFO*. The *UFO* promoter we used leads to additional expression in the rib meristem (RM) and boundary region, in addition to the PZ. Therefore, we denoted the line as *UFO'* to indicate the expanded expression domain. In addition, the *AS2* promoter we used gives similar levels of expression in young and mature leaves, whereas endogenous *AS2* has much lower expression in $P_6$ and older leaves[25]. Despite the quantitative difference, the *AS2* promoter and the endogenous gene have similar expression domains.

Domain-specific expression of HF-RPL18 can efficiently incorporate epitope tags into polysomes for immunopurification of translating cellular mRNAs from target domains[3,5]. To isolate translating mRNA from target domains, we immunopurified polysomes from seedlings from which roots had been removed at 7 days after germination (DAG). Then, we used deep sequencing to map and quantify these mRNA samples (Supplementary Fig. 2). In subsequent analysis, we also combined leaf boundary and whole leaf gene expression datasets, which were defined by the *LATERAL SUPPRESSOR* (*LAS*) and *AS1* promoters, obtained from an earlier study using identical protocols[16]. Taken together, these domains represent many of the key domains of the SAM and developing leaves.

We obtained 16.7–45.7 million uniquely mapped 50-bp reads from each of the three independent replicate libraries for each domain sample (Supplementary Table 1), which is sufficient to reliably detect rare, yet biologically relevant mRNA species of the *Arabidopsis* genome[3]. Correlation and hierarchical clustering analysis indicated that the three independent biological replicates of each domain were clustered with each other and separated from other domains (Supplementary Figs. 3, 4), suggesting good reproducibility of the translatome data.

**Domain-specific expression patterns.** To further ensure the quality and reliability of our data, we compared our translatome dataset with published data, such as in situ hybridization results. We used 20 genes with well-characterized domain-enriched expression in the SAM and/or leaves, and analyzed the enrichment levels of their encoded RNAs based on our translatome dataset. As shown in Fig. 1b, Supplementary Fig. 5 and Supplementary Data 1, we detected the expected enrichment and depletion for all transcripts, which validates the domain-specific translatome profiling data. It should be noted that *UFO* has higher enrichment in the *WUS* and *CLV3* domains in the translatome data, due to the expansion of the *UFO'* domain. There is a large number of *AS1* domain-enriched genes, which is likely due to temporal difference among *AS1*, *AS2*, and *FIL* promoters. Although spatially the *AS1* domain covers both the *AS2* and the *FIL* domains, the *AS1* promoter is only active in young leaf primordia earlier than $P_6$. In older leaves, *AS1* is only expressed in the vascular region[16,25].

The translatome of each domain was distinct, consisting of transcripts from 16,297 to 17,330 genes (49.0–52.1% of all annotated *Arabidopsis* genes, Fig. 1c). We observed that 14,152 genes (42.5%) were translated in all domains. On the other hand, a significant proportion of transcripts was detectable only in one or a few domains (Fig. 1d).

Although many genes were commonly expressed and translated in different domains, the abundance of their ribosome-bound transcripts could be highly variable among domains

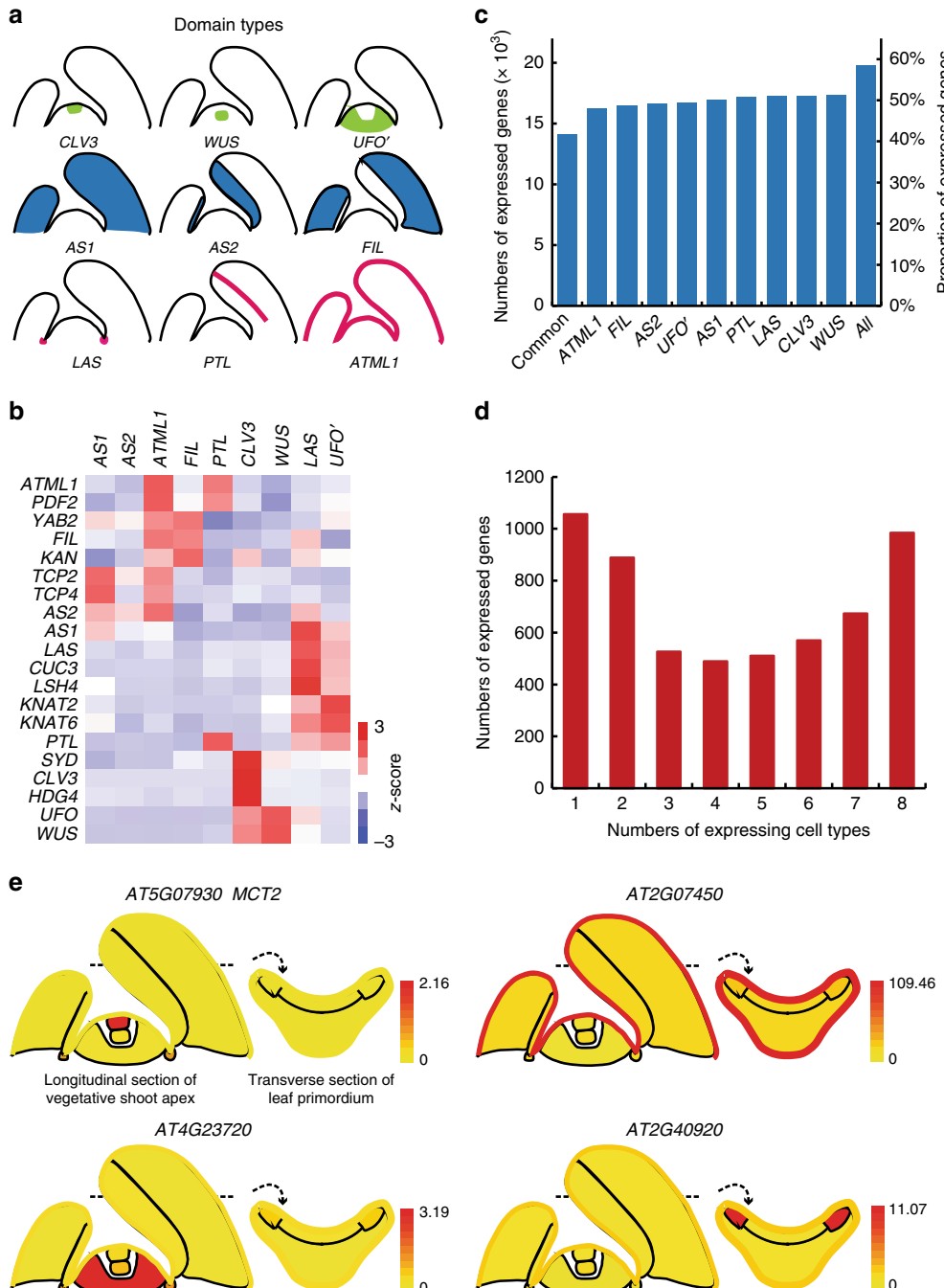

**Fig. 1** Properties of cellular gene expression. **a** Diagrams showing different spatial domains being profiled. **b** Marker gene expression pattern in different cellular transcriptomes. Relative expression (*z*-score) is displayed in colors, in which red indicates enrichment and blue indicates depletion as shown. Labels on the top show domains. The gene names are shown on the left. **c** Numbers and proportion of expressed genes in each domain. **d** Numbers of expressed genes in multiple domain groups. **e** eFP browser view of expression changes in the shoot apex for selected genes. Whereas *MEI2 C-TERMINAL RRM ONLY LIKE 2* (*MCT2, AT5G07930*) is a previously identified CZ-specific gene[21], all others are unknown domain-specific genes. Absolute gene expression values were calculated and shown for each domain

(Supplementary Fig. 6). For example, the *CLV3* domain CZ cells of the SAM are enriched in transcripts with high cell specificity (quantified by *z* values, see Methods), suggesting a dramatic translatome change during CZ to PZ transition.

To show expression dynamics graphically, we implemented an electronic fluorescent pictograph (eFP) browser[29] to show cell-specific expression data (http://bar.utoronto.ca/efp_arabidopsis/cgi-bin/efpWeb.cgi?dataSource=Shoot_Apex). Figure 1e shows a few transcripts highly enriched in selected domains.

**Domain signature genes**. In order to understand the unique cellular properties, we identified domain-specifically expressed genes. We first adopt a highly stringent pair-wise comparison method[5], in which a gene with a much higher RPKM (reads per kilobase million) value in one domain than all the other 8 domains ($\log_2FC > 1$, false discovery rate (FDR) < 0.05) is considered as domain specific. Using this criterion, we identified 1628 specific genes (Supplementary Fig. 7a, Supplementary Data 2), ranging from 19 genes in the *UFO'* domain to 420 genes

in the *AS1* domain. Biological functions associated with these domain-specifically expressed genes, identified by gene ontology (GO) analysis (Supplementary Data 3), supported data quality. For example, Photosynthesis was associated with the *AS1* and *AS2* domains, Response to Auxin Stimulus was associated with the *ATML1* domain, and Shoot Development was associated with the *LAS* domain.

We next used a modest *z*-score-based criterion, which allows retrieving genes enriched in two or a few domains. The *z*-score is calculated utilizing the mean and the standard deviation of all domains (see Methods)[4]. Domain-enriched gene lists obtained at *z*-score ≥ 2 covers the majority of the domain-specific genes obtained by the above-mentioned pair-wise comparison method (Supplementary Fig. 7b). They also have substantial overlaps with domain-enriched genes obtained by the Compartment specificity (CS) scoring algorithm[30], which compares gene expression in one cell with the maximum of the other 8 cells. We used the union of genes with *z*-score ≥ 2 and domain-specific genes identified by the pair-wise comparisons as domain-enriched genes for subsequent analysis (Supplementary Data 4). For each domain, we identified a set of domain-enriched RNAs, ranging from 150 in *UFO'* cells to 1656 in *CLV3* cells (Fig. 2a).

From the wealth of domain-enriched genes, we identified previously unrecognized cellular properties using GO analysis (Supplementary Fig. 8) which provide clues for further functional identification of these domains; for example, the categories Epigenetic Modification and Shoot Morphogenesis in the *CLV3* domain, Cell Wall Organization and Cell Wall Loosening in the *ATML1* domain, and also Glycoside Metabolic in the *UFO'* domain. We also performed hormone-response gene analysis (Supplementary Fig. 9a), enrichment of transcription factor-encoding genes (Supplementary Fig. 9b), and enrichment of DNA motifs as putative promoter *cis*-elements (Supplementary Fig. 9c) by comparison among all nine domains. It showed a remarkable diversity of these items in different domains. Similar and additional cellular properties were uncovered from more rigorous comparisons between related domains, as elaborated below.

**Similarities among different domains**. Transcriptome-wide analysis provides a unique opportunity to compare distinct domains. We first sorted to find similarities among the shoot apex domains. A principal component analysis shows tissue-cell hierarchical relationships among domain translatomes (Fig. 2b), which is also supported by hierarchical clustering analysis (Fig. 2c). SAM domains were more similar to each other, whereas leaf *FIL* and *AS2* domains group together. Neighboring domains may have similar expression profiles. On the other hand, seemingly unrelated domains can share high similarity in gene expression: *PTL*-expressing leaf margin cells are similar to *LAS* and *UFO'* cells, suggesting that the leaf marginal domain may

share meristem activities. This observation supports the proposal that parallel morphogenetic programs are shared by the leaf marginal region and the SAM[31].

**Domain-specific alternative splicing and lncRNA expression**. Alternative splicing (AS) contributes to the diversity of the transcriptome and the proteome. Studies have identified that over 60% of intron-containing genes may have AS during different developmental stages and under stress[32]. Recent studies have shown that AS events can be domain specific[33,34]. Compared with transcriptome profiling, translatome profiling provides a better estimation of the contribution of AS to proteome diversity[3,35]. In our translatome dataset, we observed substantial domain-specific AS events. A total of 4261 genes have more than one TAIR (The Arabidopsis Information Resource)-annotated isoform expressed (RPKM ≥ 1). Although splicing isoforms of the same gene generally show similar domain expression, we identified 751 genes whose isoforms showed domain-specific enrichment (*z*-score ≥ 2). For example, AT1G28330, a nuclear-enriched dormancy-associated protein-encoding gene[36], showed intron retention in the *CLV3*, *WUS*, and *PTL* domains. AT2G01180, encoding a phosphatidate phosphatase, lacked intron retention in only the *AS1*, *ATML1*, *FIL*, and *LAS* domains (Fig. 3a). Notably, there is a clear enrichment of domain-specific AS events in the *CLV3*, *WUS*, and *LAS* domains (Fig. 3b), suggesting roles of AS in the specific activities of these domains.

In addition to protein coding genes, long noncoding RNAs (lncRNA) have gained attention in recent years[34,37]. Although lncRNA in general lacks protein coding capacity, we have previously found that some of them may associate with polysomes[3], implying potential roles in regulation of translation, or that their characterization as lncRNAs ignores protein coding potential. Ribosome-bound lncRNAs were also found in animals and *Arabidopsis*[38–40]. Using recent expression-based lncRNA annotations[34,41], as well as de novo assembly based on our own data, we analyzed domain-specific lncRNA distribution in the translatome dataset. Together, we detected 242 lncRNAs in one or more domains (Supplementary Data 5). Among them, 13 are previously unknown lncRNAs, and 21 overlap with but extend previous annotations at the 5' or/and the 3' ends. We found 117 lncRNAs (48.3% of all expressed ones) were enriched in one domain (Fig. 3c), which is substantially higher than the proportion for mRNAs. The *WUS* domain and leaf domains share distinct sets of enriched lncRNAs, suggesting possible involvement of lncRNAs in development.

Besides lncRNAs, we also found 125 pseudogenes with moderate expression levels in the translatome datasets. Among them, 40 pseudogenes showed a domain-specific expression pattern (Fig. 3c, Supplementary Data 6). The biological roles of the polysome-associated lncRNAs and pseudogenes warrant

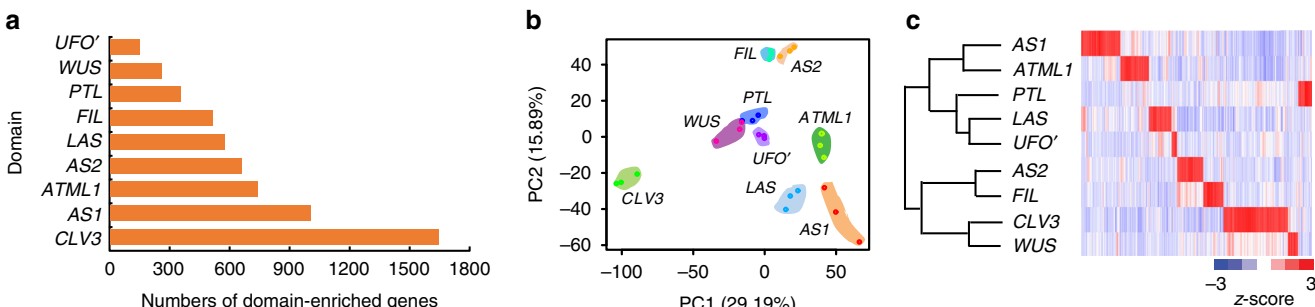

**Fig. 2** Domain-specific expression patterns. **a** Number of domain-enriched genes in each domain. **b** Principal component analysis of the nine domains with triple biological repeats for each type. **c** Hierarchical clustering analysis of domain-enriched genes. Complete linkage hierarchical clustering was performed on relative expression (*z*-score) as described in Methods

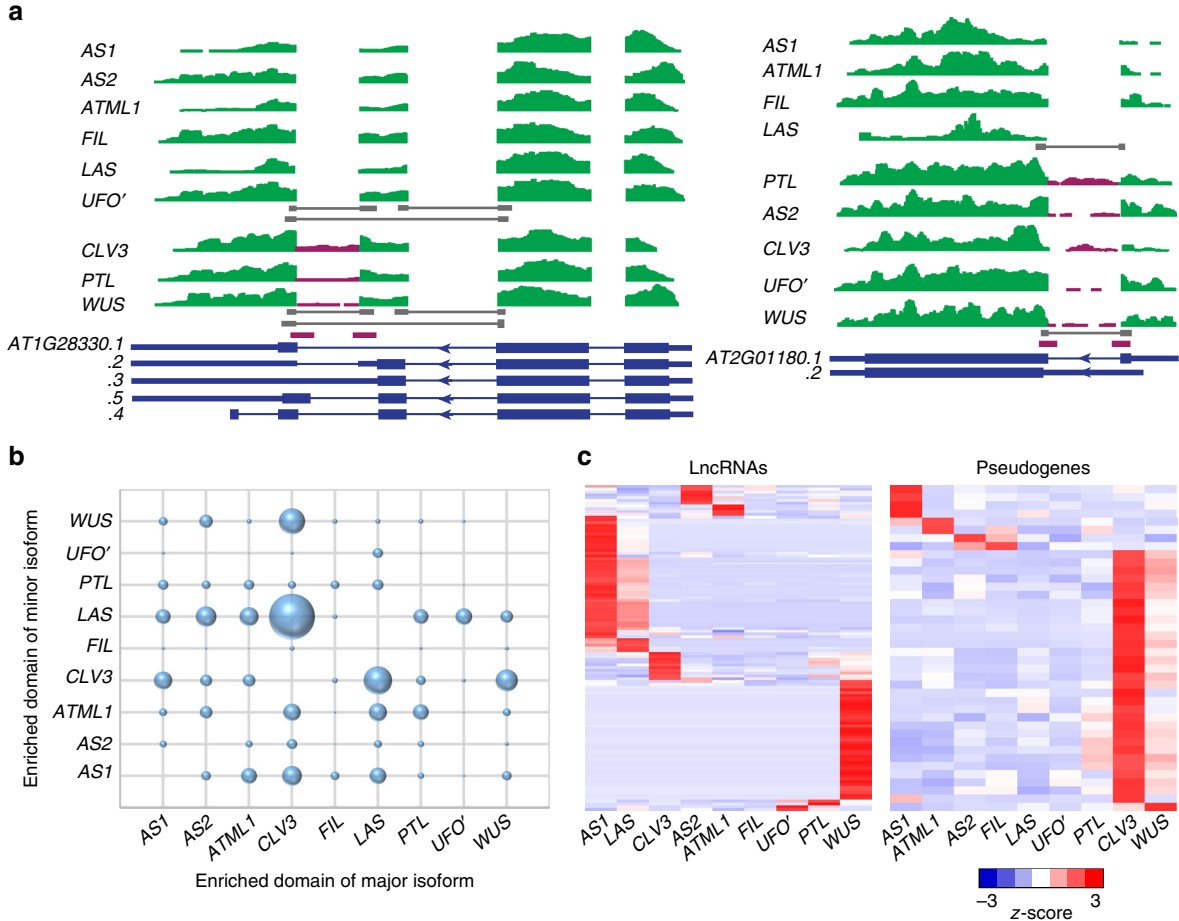

**Fig. 3** Domain-specific expression of AS and lncRNAs. **a** Examples of domain-specific expression of alternative splicing. Reads distribution of two genes are given. TAIR10-annotated gene models (i.e., isoforms) for each locus are represented as blue boxes in the bottom. Areas in purple color indicate reads which were differentially distributed among isoforms, while those in green are common ones. Selected reads covering exon–exon junctions are highlighted by gray bars. Purple boxes represent reads containing exon–intron junction sequences in some isoforms. **b** Enriched domain of alternative splicing. The size of the balls represents numbers of enriched AS genes. **c** Expression pattern of domain-enriched long noncoding RNAs and pseudogenes. Complete linkage hierarchical clustering was performed on relative expression (z-score) as described in the Methods

further investigation. Whether these lncRNAs and pseudogenes harbor hidden open reading frames should also be determined.

**RBE and HAN regulate axillary meristem initiation**. Utilizing the translatome data of different shoot apex domains, we performed a gene co-expression network analysis (GCN)[42,43]. A total of 7085 expressed genes with high coefficient of variation ( > 0.7) fell into18 subnetwork GCN modules (M1 to M18). Each module includes from 72 (M11) to 1331 (M2) co-expressed genes (Supplemental Fig. 10). When compared with domain-enriched genes, 17 out 18 GCN modules displayed strong associations with one or more domains (hypergeometric test with a FDR correction, FDR < 0.01, Fig. 4a). Some modules overlapped with two domains, for instance, *CLV3* and *WUS* in module 2, *LAS* and *UFO'* in module 5, which may suggest the similar regulatory mechanisms in those two domains (Fig. 4a). Because transcription factors play central roles in gene regulation, we further investigated transcription factors in each module, which contained different numbers of co-expressed transcription factors, from 3 (M16) to 126 (M8) (Supplemental Fig. 10).

Domain co-expression analysis provides insights into potential gene functions. Among these modules, we found module M12 was enriched with known boundary-specific genes, such as *LAS*, *CUP-SHAPED COTELYDON*s (*CUCs*), *REGULATOR OF*

*AXILLARY MERISTEMS1*, and *LATERAL ORGAN FUSION1* (Fig. 4b, Supplementary Data 7). *ARABIDOPSIS THALIANA HOMEOBOX PROTEIN29* and *PTL* within this model are upstream regulators of *LAS* and *CUC2*, respectively[16]. *RABBIT EARS* (*RBE*), which acts in flower development[44], was also found in this module, implying a possible previously unknown function in vegetative shoot development. *RBE* encodes a C2H2 family zinc finger transcriptional repressor, whose loss-of-function mutants exhibit aberrant petals and fused sepals[44,45]. *RBE* is enriched in the boundary domain (Fig. 4c). We found ectopically enhanced AM initiation in loss-of-function *rbe-2* mutant plants (Fig. 4d, Supplementary Fig. 11a). Whereas accessary buds, which form between axillary buds/branches and leaf petioles, are rarely seen in wild-type plants, they are reproducibly found in *rbe-2* plants at high frequency (Fig. 4e).

In another module, M5, we also found enrichment of boundary-specific genes (Fig. 5a, Supplementary Data 8). A number of boundary-specific LBD and ALOG transcription factors fell into this module. Among genes without known boundary function, *HANABA TARANU* (*HAN*, also known as *MONOPOLE* and *GATA18*), encoding a GATA transcription factor, is highly expressed in both the leaf axil and the OC (Fig. 5b). Previous studies have shown its function in the SAM, in flower development, and in embryogenesis[46–50]. The strong expression of *HAN* in the leaf axil implies as-yet unknown

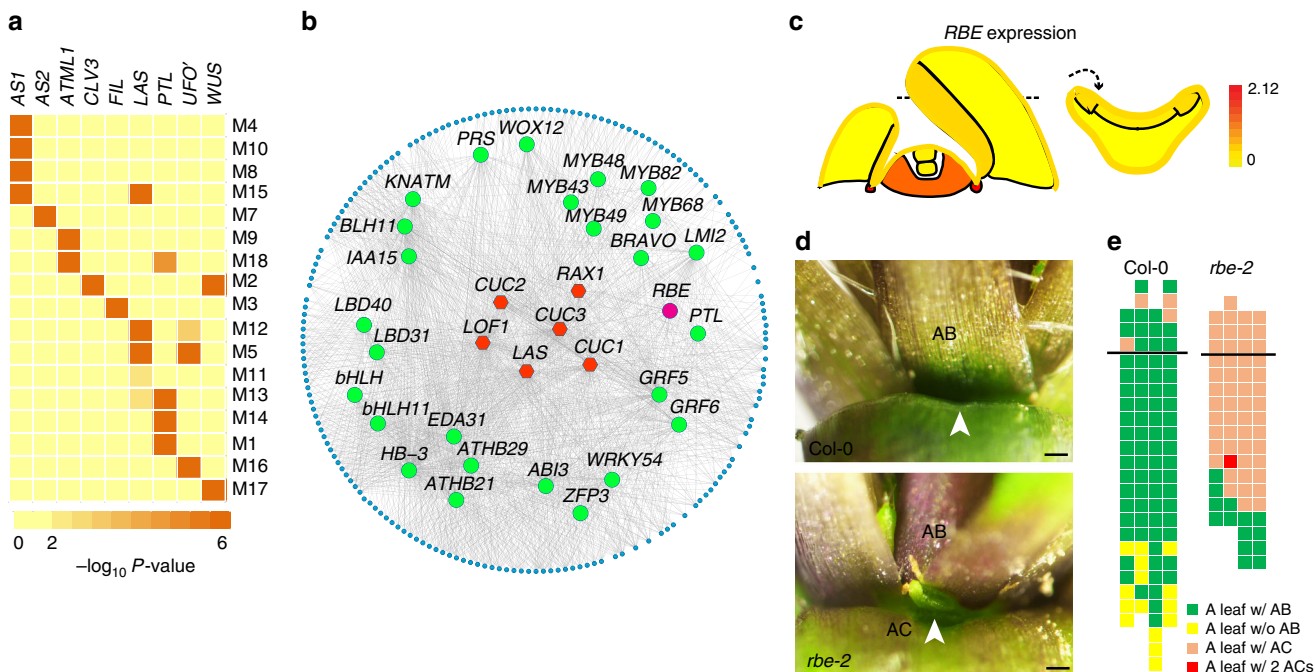

**Fig. 4** Boundary-specific module and function of *RBE* in AM initiation. **a** Relationships of the domain-enriched genes with GCN modules. Colors indicate the enrichment assessed by hypergeometric tests of overlapping gene numbers. Labels on the top show domain and corresponding modules displayed in the right panel. **b** Module M12 displayed by Cytoscape showed a significant enrichment of known boundary-specific genes involved in AM initiation. Transcription factors are in the middle, in which red color indicates known AM regulators, while green represents others. *RBE* is represented in purple. Dots around the circle represent other co-expressed genes. Lines represent relationships. **c** Expression pattern of *RBE* viewed by the eFP browser. **d** Increased AM initiation in *rbe-2* mutant. Rosette leaf axil from Col-0 with a single axillary branch (AB), and from *rbe-2* mutant with an accessory bud (AC, arrowhead). Scale bar = 200 μm. **e** Schematic representation of axillary bud formation in leaf axils of wide-type and mutant plants. Each column represents a single plant, and each square within a column represents an individual leaf axil. The horizontal line indicates a border between cauline leaf and rosette leaf, with the youngest to oldest from top to bottom. Green or yellow colors indicate the presence or absence of an axillary bud in the particular leaf axil. Orange or red color indicates the presence of one or two accessory buds, respectively

biological functions. To this end, we analyzed AM initiation phenotypes using *han* mutants. We found clear AM initiation defects in both the *han-2* and *han-30* mutants (Fig. 5c, d), in which AMs could no longer initiate in ~54.6% of rosette leaves, and also in cauline leaves. Thus, *HAN* has a role in promoting AM initiation, which is consistent with its enriched expression in the leaf axil. The mutation sites of *han-2* and *han-30* are located in the zinc finger domain and HAN motif, respectively (Supplementary Fig. 11b), indicating roles of these two domains in AM initiation.

**Comparison of gene expression trends among SAM domains**. The three SAM domains, *CLV3*, *WUS*, and *UFO'*, have highly specialized gene expression, which may reflect functional specialization in each domain. A restricted comparison among these SAM domains identified a substantial number of genes specific for each domain, with the *UFO'* domain more distinct from the *CLV3* and *WUS* domains (Fig. 6a). In total, we detected transcripts of 1079 genes, corresponding to 5.8% of all expressed genes, as *UFO'* domain specific. There were 512 and 423 (2.8% and 2.3%, respectively) *CLV3* and *WUS* domain-specific RNAs (Supplementary Data 9). The large number of *UFO'* domain-specific transcripts may correspond to active cell proliferation in this domain. Significant enrichment of phytohormone responses and transcriptional activity also suggest the *UFO'* domain-specific gene regulatory network extensively uses phytohormones (Fig. 6b).

GO analysis also provides genome-level support to recently identified physiological functions of each domain (Fig. 6c). Notably, we observed Cell Wall Loosening and Lignin and

Flavonoid categories were significantly enriched in the *UFO'* domain, consistent with recently identified roles of cell wall mechanical stress changes in this domain[51–53]. Moreover, enrichment of many GO categories suggests previously uncharacterized localized physiological functions. For example, Lipid localization and Response to Stress categories were enriched in the *UFO'* domain, implying their active involvement in leaf initiation. Notably, transcripts related to photosynthesis were enriched in the *WUS* domain. Although we do not yet know its biological implication, a similar enrichment was observed in the *WUS* domain of the FM[20], and also during shoot regeneration from lateral root primordia[54].

**Gene regulation in the epidermis**. The shoot epidermis is a single layer of cells that covers the plant body. The epidermal cells are almost exclusively derived from anticlinal cell divisions so that the entire epidermal layer is generated from the L1 layer of the SAM[55]. Experimental and theoretical studies have shown that the epidermis has distinct wall properties to restrict internal growth, and to control organ patterning and size[56–59]. The epidermis of seed plants generates the cuticle and cuticle wax as additional specializations that cover the outer surface of plants. To explore the physiological functions of the epidermis, we compared the epidermis, defined by *ATML1* promoter activity, with leaf cells, defined by *AS1* promoter action.

The epidermis is highly distinct from inner cells with 1471 enriched and 2180 depleted classes of transcripts, corresponding to 8.3 and 12.3% of all expressed genes (Supplementary Fig. 12a, Supplementary Data 10). Based on these dominant expression signatures, we uncovered many over-represented GO terms

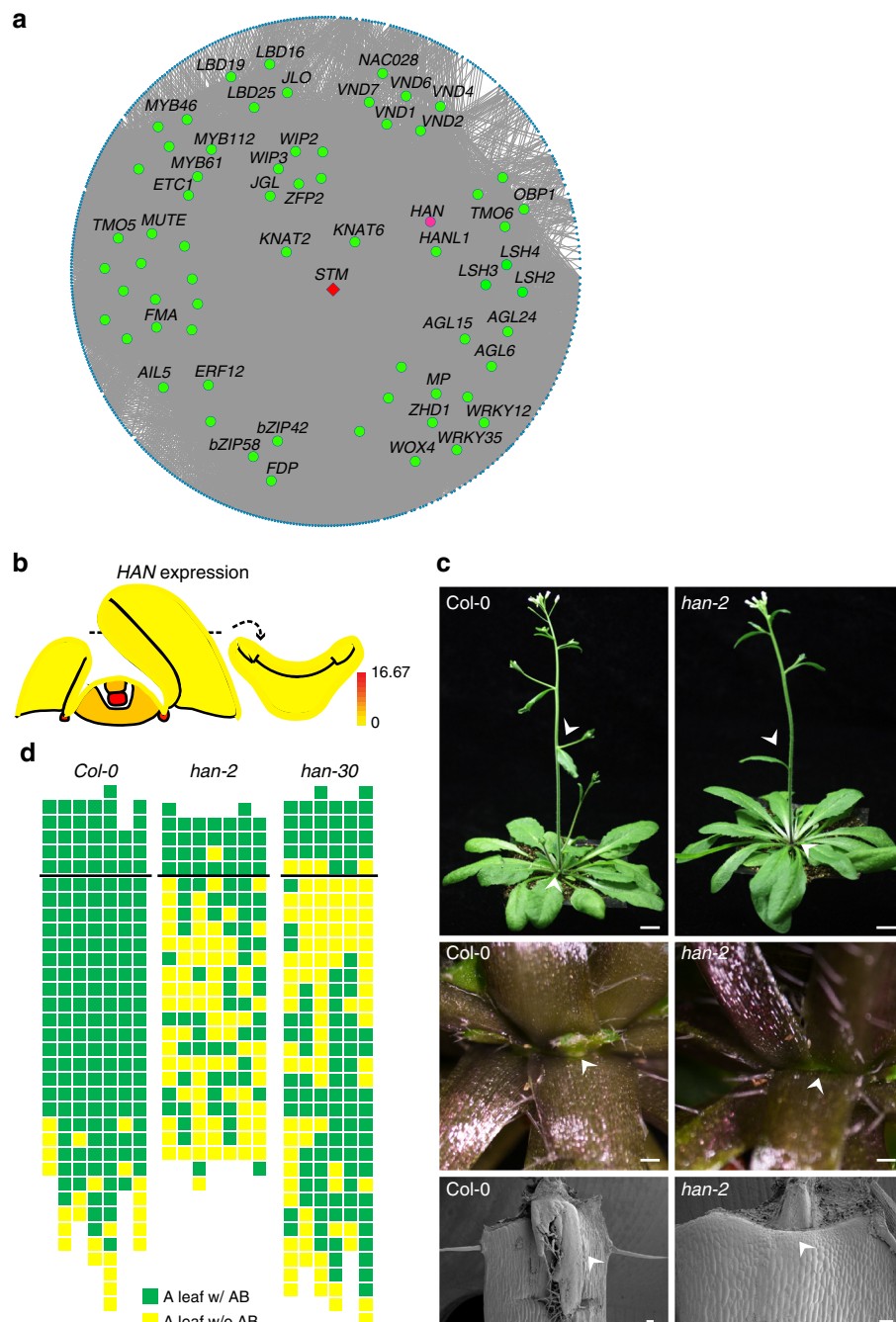

**Fig. 5** GCN module M5 and function of *HAN* in AM initiation. **a** Module M5 displayed by Cytoscape as in Fig. 4b. Transcription factors are in the middle, with known AM regulators in red, *HAN* in purple, and others in green. **b** Expression pattern of *HAN* viewed by the eFP browser. **c** AM defects in *han-2* mutants. The whole plant (upper panel, scale bar = 1 cm), close-up of rosette leaf axils (middle panel, scale bar = 500 µm), and also scanning electron microscope view (lower panel, scale bar = 100 µm) in Col-0 and *han-2* mutant plants showing the presence and absence of AMs (arrowhead), respectively. **d** Schematic representation of axillary bud formation in leaf axils of wide-type and mutant plants. Each column represents a single plant, and each square within a column represents an individual leaf axil. The horizontal line indicates a border between cauline leaf and rosette leaf, with the youngest to oldest from above to bottom. Green or yellow color indicates the presence or absence of an axillary bud in the particular leaf axil, respectively

(Supplementary Fig. 12b). For example, RNAs associated with Epidermis Cell Differentiation, Wax Biosynthesis, and Cell Wall Organization were all significantly enriched in the epidermis, confirming known physiological functions. As another example, genes encoding extensins and expansins were also enriched in the epidermis (Table 1), as might be expected if the expansion of epidermal cells controls that of inner cell layers[56,59]. In addition, genes involved in trichome and stoma formation are highly enriched (Table 2). Also, "Response to Auxin Stimulus" was enriched, consistent with the known active auxin transport and distribution in the epidermis, and this category included transcripts from AUX/IAA family genes (Supplementary Fig. 12c). In addition, we observed that RNAs of genes whose expression responds either positively or negatively to other hormone stimuli were enriched in the epidermis, suggesting that epidermal function involves active hormonal signaling functions and integration (Supplementary Fig. 12d).

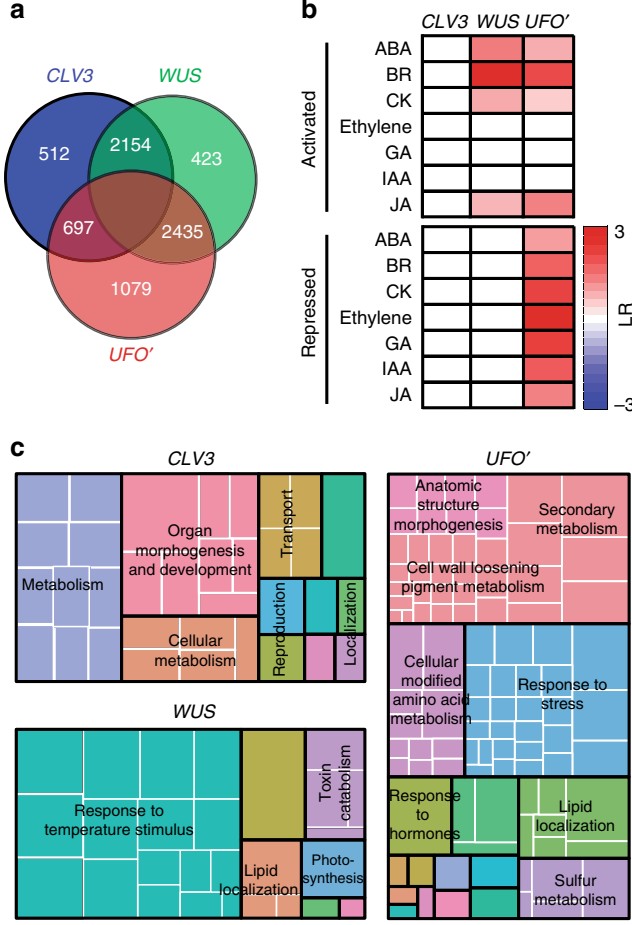

**Table 1 Plant cell wall-related genes enriched in epidermal cells**

| Gene ID | Name | Log₂FC | FDR |
|---|---|---|---|
| AT2G24980 | *EXTENSIN 6* | 3.91 | 8.64E−05 |
| AT5G06640 | *EXTENSIN 10* | 4.53 | 3.62E−05 |
| AT4G13390 | *EXTENSIN 12* | 7.37 | 1.23E−08 |
| AT5G35190 | *EXTENSIN 13* | 7.52 | 1.11E−06 |
| AT3G28550 | *EXTENSIN* | 6.05 | 9.74E−12 |
| AT4G08400 | *EXTENSIN* | 7.85 | 1.59E−05 |
| AT3G54580 | *EXTENSIN* | 5.72 | 3.27E−08 |
| AT1G23720 | *EXTENSIN* | 5.29 | 1.45E−07 |
| AT3G29030 | *EXPANSIN A5* | 1.72 | 4.29E−06 |
| AT2G40610 | *EXPANSIN A8* | 2.102 | 2.38E−14 |
| AT1G20190 | *EXPANSIN A11* | 2.90 | 3.96E−14 |
| AT3G55500 | *EXPANSIN A16* | 2.70 | 4.71E−04 |
| AT1G62980 | *EXPANSIN A18* | 4.39 | 1.40E−03 |
| AT3G45960 | *EXPANSIN-LIKE A3* | 2.40 | 1.69E−07 |
| AT1G10550 | *XYLOGLUCOSYL TRANSFERASE 33* | 2.61 | 2.04E−06 |
| AT2G21140 | *PROLINE-RICH PROTEIN 2* | 1.74 | 8.34E−06 |
| AT1G31420 | *FEI1* | 1.23 | 4.31E−08 |

**Fig. 6** Gene expression pattern among SAM domains. **a** Differentially expressed genes among different domains in the SAM. All differentially expressed genes between any two meristem domains with FC ≥ 2, FDR ≤ 0.01 were used for the analysis. Domain-specific genes are those enriched when compared with the other two domains. **b** Domain-specific enrichment of hormone-activated and -repressed genes among the three SAM domains. Only significantly over-represented hormone categories by a hypergeometric test with a FDR correction (adjusted *P* value ≤ 0.05) are colored with log₂ scaled odds ratio (LR). **c** GO analysis of domain-specific genes, which indicates function-related enrichment in different domains summarized using REVIGO. Aggregate size indicates significance levels in the GO term by the Yekutieli test with FDR correction

**Abaxial domain as a responsive center.** Soon after initiation from the SAM, leaf primordia develop dorsiventral (adaxial–abaxial), proximodistal, and mediolateral polarity patterns. These polarities are necessary for leaf laminar expansion and leaf domain specification[13].

We compared the gene expression profiles in the adaxial domain, defined by *AS2*, and the abaxial domain, defined by *FIL*. The comparison led to the identification of 915 adaxially enriched and 1077 abaxially enriched transcripts, corresponding to 5.3 and 6.3% of all expressed genes (Supplementary Fig. 13a, Supplementary Data 11). We identified many physiological functions in the abaxial domain. GO enrichment analysis indicated that many biotic and abiotic stress responses and responses to phytohormones were active in the leaf abaxial domain (Fig. 7). One the other hand, the adaxial domain was enriched with photosynthesis functions (Supplementary Table 2).

We further focused on phytohormone responses. Based on previously identified phytohormone-responsive genes[16], we found that transcripts from genes whose expression is activated

by a phytohormone, including abscisic acid, auxin, brassinosteroids, ethylene, gibberellin, and jasmonic acid, were enriched in the abaxial domain. On the other hand, transcripts from genes whose expression is repressed by abscisic acid, brassinosteroid, cytokinin, gibberellin, and jasmonic acid were also enriched in the abaxial domain, and those repressed by auxin and ethylene were enriched in the adaxial domain (Supplementary Fig. 13b). This genome-wide observation corresponds with recently reported hormone signaling activities in leaf primordia[17]. Additional domain-specific phytohormone signaling suggests possible roles for ethylene, gibberellin, and jasmonic acid. Correspondingly, RNAs from some specific transcription factor families and promoter motifs were also enriched in the abaxial domain (Supplementary Fig. 13c, d).

## Discussion

In this study, we present a map of rigorously comparable shoot domain-specific translatome profiles. A total of 19,850 genes, corresponding to 59.7% of all annotated *Arabidopsis* genes, were ribosome associated in at least one shoot domain (Fig. 1c). Although most genes (14,152 genes) were detected in all domains, many domain-enriched and domain-specific genes were identified. A systematic dissection of gene regulatory networks requires comprehensive, precise, yet rigorous gene expression data. This study offers a platform for a broad range of studies to understand the specialized functions of shoot domains.

A tissue or gene expression domain resolution expression map can aid in the dissection and validation of gene regulatory networks by providing evidence of the co-expression of potential pathway members within each domain. Such information would not be evident in data obtained from whole organs. For instance, we inferred roles for *RBE* and *HAN* in the leaf axil from co-expression analysis, and experimentally identified AM initiation changes in *rbe* and *han* mutants (Figs. 4, 5).

More focused comparison among a subset of cell domains may reveal further insights into domain functions. By analyzing SAM cells, shoot epidermal cells, and leaf adaxial and abaxial cells, we provide genome-wide support for recently identified biological roles of these domains, such as wall stiffness control and phytohormone signaling. Our genome-wide analysis also leads to insights into these domains (Figs. 6, 7). Some of these unknown processes are unexpected, such as the enrichment of photosynthesis-associated genes in the *WUS* domain. Many of

**Table 2 Epidermal cell differentiation-related genes enriched in epidermal cells**

| Gene ID | Name | Log$_2$FC | FDR |
|---|---|---|---|
| AT2G38120 | *AUXIN RESISTANT 1* | 1.15 | 9.41E−05 |
| AT4G01060 | *ENHANCER OF TRY AND CPC 3* | 3.68 | 2.04E−06 |
| AT2G41940 | *ZINC FINGER PROTEIN 8* | 1.21 | 4.51E−05 |
| AT2G26250 | *3-KETOACYL-COA SYNTHASE 10* | 2.29 | 3.40E−30 |
| AT1G12040 | *LEUCINE-RICH REPEAT/EXTENSIN 1* | 6.91 | 7.90E−09 |
| AT4G21750 | *MERISTEM LAYER 1* | 1.40 | 5.67E−11 |
| AT1G56580 | *SMALLER WITH VARIABLE BRANCHES* | 2.52 | 3.06E−19 |
| AT3G24140 | *FAMA* | 2.38 | 4.55E−15 |
| AT1G05230 | *HOMEODOMAIN GLABROUS 2* | 1.56 | 4.23E−08 |
| AT1G14350 | *FOUR LIPS* | 1.63 | 1.02E−09 |
| AT3G62680 | *PROLINE-RICH PROTEIN 3* | 5.16 | 8.14E−05 |
| AT1G73360 | *HOMEODOMAIN GLABROUS 11* | 1.48 | 7.83E−03 |
| AT1G17920 | *HOMEODOMAIN GLABROUS 12* | 2.44 | 2.72E−09 |
| AT2G22640 | *BRICK1* | 1.52 | 1.63E−05 |
| AT4G04890 | *PROTODERMAL FACTOR 2* | 2.08 | 7.22E−25 |
| AT2G47000 | *ATP-BINDING CASSETTE B4* | 1.70 | 1.64E−04 |
| AT2G26650 | *K+TRANSPORTER 1* | 1.41 | 2.26E−05 |
| AT5G55480 | *SHV3-LIKE 1* | 1.42 | 3.46E−08 |

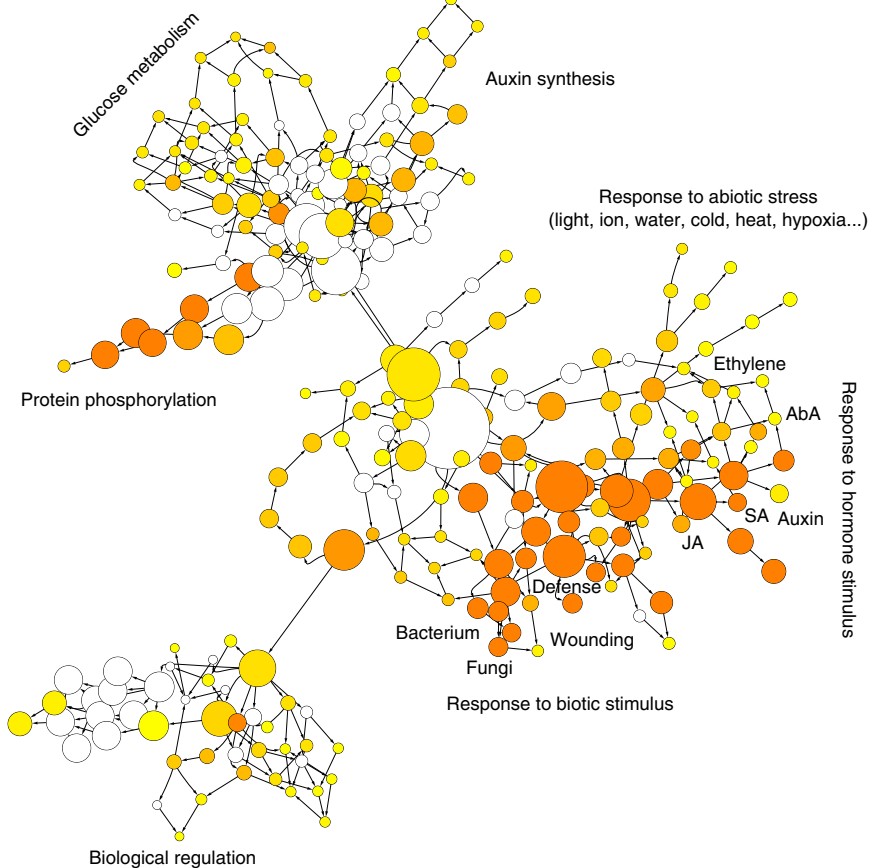

**Fig. 7** Dominant patterns of gene expression in leaf abaxial domain. GO enrichment shows that leaf abaxial domain is a stimulus- and phytohormones-responsive center. The regulatory relationship was displayed by Cytoscape

these unknown processes are intriguing. For example, the response to temperature stimulus in the *WUS* domain might represent a regulatory module that connects environmental temperature to stem cell proliferation and overall growth. In addition, our analysis uncovered previously unrecognized similarities among seemingly unrelated domains (Fig. 2b, c).

Compared with transcriptome analysis, translatome analysis provides additional insights into post-transcriptional regulation.

By comparing transcriptome and translatome profiles, it has been shown that a substantial portion of the transcripts are under translational regulation[3,60]. Whereas deep proteomic profiling remains a challenge, the translatome serves as a more convenient proxy for proteomic studies. In addition, translatome profiling can experimentally annotate ncRNAs that are often overlooked in bioinformatic annotation pipelines[39,40]. We observed a substantial proportion of lncRNAs to be associated with polysomes,

suggesting their involvement in translation or some related process. Some lncRNAs are enriched in certain domains. Ribosome profiling that combines ribosome footprinting with deep sequencing can provide additional insights into small upstream open reading frames (ORFs) and translation pausing[40,61,62]. On the other hand, the TRAP-seq (translating ribosome affinity purification followed by sequencing) approach, which does not include ribosome footprinting, can reveal AS events that lead to proteome diversity. In this analysis, we identified domain-specific AS events, reflecting differential contribution of selected AS events during shoot domain specification.

In summary, this domain-specific translatome map covers many of the major cell types of a shoot. Together with the accessible pictograph browser, it should provide a valuable community resource for further investigations of shoot development. It will also provide a starting point to understand responses to environmental and internal signals at domain resolution. As datasets for more domains and conditions are added, it should be possible to distinguish a characteristic expression fingerprint for each domain, which will yield further insight into the nature of domains themselves.

## Methods

**Plant materials and generation of transgenic plants**. The *Arabidopsis thaliana* accession Landsberg *erecta* (L*er*) was used as the wild type for TRAP-seq lines unless otherwise specified. Information on the detailed genetic background of mutants and transgenic lines used in this study is provided in Supplementary Table 3. A *han-2* line from a mixed L*er* and Col-0 background (without the *er* mutation)[46], *han-30* in Col-0[49], and *rbe-2* (SALK_037010) in Col-0[63] were used for phenotypic analyses. Supplementary Figure 11 summarizes their mutation sites. The genotypes of all mutants were verified by PCR amplification and sequencing. For *pUFO::LhG4* and *pAS2::LhG4*, 5.9 kb upstream sequence of *UFO*[22] and 3.3 kb upstream sequence with additional 18 bp of the N-terminal *AS2* coding region[25,64] were cloned adjacent to the coding sequence of the LhG4 protein into the BJ36 vector and the fragments *pUFO::LhG4* and *pAS2::LhG4* were subcloned into the pMLBart vector, then transformed into L*er*. To identify the expression pattern of each promoter, we crossed domain-specific LhG4 drivers into a *pOp::GFP-ER* driver line. Lines with correct expression patterns were then crossed into a *pOp::HF-RPL18* driver line which also contains a linked *pOp::GUS* for further profiling (Supplementary Fig. 1). For *pCLV3::HF-RPL18*, a 1.5 kb sequence upstream and 1.2 kb downstream of the *CLV3* coding sequence[65] were directly cloned before and after the *HF-RPL18* sequence. The *pCLV3::HF-RPL18:tCLV3* fragment was then subcloned into pMLBart and transformed into L*er*. Transgenic lines with correct CZ-specific HF-RPL18 expression were selected by immunolocalization (Supplementary Fig. 1). Primers used are listed in Supplementary Table 4.

**TRAP-seq**. Seedlings grown on 1/2 MS agar plates containing 1% sucrose were used at 7 DAG. For translatome profiling, shoots were frozen in liquid nitrogen, powdered with mortar and pestle, and homogenized in ice-cold polysome extraction buffer (200 mM Tris-HCl (pH = 9.0), 200 mM KCl, 36 mM MgCl₂, 25 mM ethylene glycol tetraacetic acid, 1 mM dithiothreitol, 50 μg/ml cyclohex-imide, 50 μg/ml chloramphenicol, 1% Igepal CA-630, 1% Brig-35, 1% Triton X-100, 1% Tween-20, 2% polyoxyethylene (10) tridecyl ether, 1% sodium deoxy-cholate, 0.5 mg/ml heparin, and recombinant RNase inhibitor). After incubation for 10 min on ice, homogenates were centrifuged at 16,000 × g and 4 °C for 10 min to pellet insoluble cell debris. Affinity purification of HF-RPL18-containing poly-somes was carried out using anti-FLAG beads (Sigma-Aldrich) incubated with the supernatant at 4 °C overnight. Gels were subsequently collected by centrifugation, and washed three times with polysome wash buffer (200 mM Tris-HCl (pH = 9.0), 200 mM KCl, 36 mM MgCl₂, 25 mM ethylene glycol tetraacetic acid, 5 mM dithiothreitol, 50 μg/ml cycloheximide, 50 μg/ml chloramphenicol, and RNase inhibitor). Polysomes were eluted by resuspension of the washed gel in wash buffer containing 3× FLAG peptide. Total RNA was extracted from the final elution using an RNeasy Kit (Qiagen) (Supplementary Fig. 2)[3,16,66]. Total RNA and subsequent poly(A)⁺ RNA were isolated from each replicate, and subjected to RNA-sequencing library preparation[3,67]. Libraries were sequenced as 50-mers using HiSeq2000 (Illumina, San Diego, CA) with standard settings. Three independent biological replicates were included for each domain.

**Read mapping and quantification of expression**. Reads were mapped to the *Arabidopsis* Information Resource TAIR10 reference genome build with TopHat2 (version 2.0.9) and BOWTIE (version 2.1.0) allowing up to two mismatches[68] after filtering the low-quality reads (PHRED quality score < 20). The gene locus expression levels were calculated based on mapping outputs after removing reads mapped to ribosomal RNAs and transfer RNAs using Cuffdiff2 (version 2.1.1)[69],

and expression levels were normalized to the RPKM unit using edgeR[70] with significant expression cutoff value set to RPKM > 1. Differential expression was assessed with edgeR and the cutoff value was >2-fold change in expression with Benjamini–Hochberg adjusted FDR < 0.01. We used Cuffdiff2 to quantify the abundance of annotated isoforms. For the identification of domain-specifically expressed genes, three methods were performed independently. Pair-wise comparison was carried out using the differential expression assessed with edgeR. One gene was identified as a domain-specifically expressed gene when it was differentially expressed from all the other eight domains (log₂FC > 1, P < 0.05). We also converted RPKM values for all samples into z-scores as relative expression levels[4]. The z-score for the *i*th gene in the *j*th domain is determined by the equation $z_{ij} = (x_{ij} - \mu_i)/\sigma_i$, where x is the expression value, and μ and σ are the mean and standard deviation in all samples. Domain-enriched genes were identified as genes with a z-score above two in a particular domain. A CS score was calculated according to a reported algorithm[30]. For a given gene *i*, its expression values in 9 domains were denoted as $EV_i = (EV^i_1, EV^i_2, ..., EV^i_9)$. So the CS score of this gene in the *j*th domain was calculated as $CS\,(i, j) = 1 - \max(E^i_k/E^i_j)$, where $1 \le k \le 9$, $k \ne j$. A gene with a CS score above 0.3 was denoted to be cell-type specifically expressed.

**Gene ontology enrichment and promoter motif analysis**. GO term enrichment analysis was performed using agriGO with the Singular Enrichment Analysis method[71] and summarized using REVIGO[72]. For the enrichment analysis of leaf abaxial differential genes, BiNGO[73] was carried out and the result was displayed by Cytoscape[74]. Lists of the phytohormone-responsive genes (Supplementary Data 12) and transcription factor classification (Supplementary Data 13) were given[16]. The gene enrichment analysis was quantified by log₂ odds ratio (LR)[3]. Briefly, to determine which categories of hormone-responsive genes or transcription factor genes (HT) are enriched with domain characteristic (DC) genes, the number of DC genes contained in each HT category was counted. LR was then calculated to qualify enrichment. $LR = \log2\left(\frac{q/k}{m/t}\right)$, where q is the count of DC genes in an HT category, k is the total number of DC genes, m is the total number of an HF category, and t is the total number of expressed genes. A hypergeometric distribution was used to assess the statistical significance (P value) of the enrichment.

Promoter motif enrichment was analyzed[3,75]. The genome sequences 2 kb upstream from annotated translation start sites for domain-specific genes were retrieved from the TAIR10 genome build to identify over-represented known sequence motifs using an enumerative approach with Elefinder (http://stan.cropsci.uiuc.edu/tools.php). Those elements meeting an expected (E) value smaller than 10⁻⁴ were selected for further comparison.

**Multivariate statistics and visualization**. Histograms of expression dynamics were produced using R packages. Hierarchical clustering analysis for marker gene and domain-enriched genes were performed in Cluster 3.0 and shown with Treeview[76]. The complete hierarchical clustering analysis for all expressed genes in each library was carried out by hclust and principal component analysis of domain-specific genes was performed by prcomp within R packages.

**Co-expression network analysis**. Co-expression network analysis was performed to identify modules of highly correlated genes using a R package WGCNA[42,43]. Genes with low coefficient of variation among domains (CV = STD/Mean, CV < 0.7) were filtered out. Finally, log₂ transformed RPKM values of 7085 genes were used to construct the network. The soft threshold power was set to 9, according to assessment of scale-free topology and a dynamic tree cutoff 0.20 was employed to merge similar trees (Supplement Fig. 14). The statistical significances of enrichment of domain enriched genes in each module were assessed by hypergeometric distribution. Co-expression networks were displayed using Cytoscape[74].

**lncRNA and pseudogene expression analysis**. We combined previous expression-based lncRNA annotations, including those obtained from TAIR10 and RepTAS databases, as known lncRNAs[34,41,77]. We also identified novel lncRNAs based on our expression data following published protocols[34,41]. In brief, we employed Cufflinks (v2.2.1)[78] and Stringtie (v1.3.5)[79] to assemble putative transcripts for each biological replicate. Meta-assemblies were performed with Cuff-merge (v1.0.0) to construct a final unified set of transcripts. Transcripts that overlapped with annotated genes were removed. Those overlapped with known lncRNAs either with extended 5' or 3' end were classified as known (Supplementary Data 5). The remaining assembled transcripts were considered novel lncRNAs if they fit the following criteria[34,41]: (1) longer than 200 bp; (2) 500 bp away from protein coding genes; (3) do not overlap with transposons; and (4) containing no ORF (predicted by webAUGUSTUS) longer than 300 bp. Pseudo-gene annotation was following TAIR10. The expression of lncRNAs and pseudo-genes were also assessed and summarized in RPKM with edgeR. The differential expression patterns were identified by z-score as for protein coding genes.

**Optical and scanning electron microscopy**. For immunolocalization, shoot apices were fixed in fresh FAA solution (3.7% formaldehyde, 50% ethanol, and 5% acetic acid) under vacuum and embedded in Steedman's wax composed of poly-ethylene glycol 400 distearate and 1-hexadecanol (Sigma-Aldrich). After rehydration, 8 μm sections were pretreated 1 h with 2% bovine serum albumin (BSA) in

phosphate-buffered saline (PBS) and incubated overnight with the anti-FLAG antiserum (Sigma-Aldrich) diluted 1:500 in PBS containing 0.1% BSA. After three washes in PBS with 0.1% (v/v) Tween-20, sections were incubated for 1 h with the secondary antibodies Alexa Fluor 488 donkey anti-mouse IgG (Life Technologies) diluted 1:1000 in PBS supplemented with 0.1% (w/v) BSA. After additional rinses in PBS plus 0.1% Tween-20, sections were mounted in ProLong Antifade (Thermo-Fisher) under cover slips and examined using a confocal laser scanning microscope. Confocal images were taken with a Nikon A1⁺ confocal microscope. A 488 nm laser line was used for excitation, and a 505–550 nm band-pass filter was used for Alexa Fluor 488, a 585–615 nm band-pass filter was used for propidium iodide, and a 660–700 nm band-pass filter was used for autofluorescence. Optical photographs were taken with a Nikon SMZ1000 stereoscopic microscope or an Olympus BX60 microscope equipped with a Nikon DS-Ri1 camera head. Scanning electron microscopy was performed to observe the fine structure of leaf axils using a Hitachi S-3000N variable pressure scanning electron microscope after standard tissue preparation.

## Data availability

TRAP-seq data have been deposited in the NCBI Short Read Archive with accession number SRP145572. The expression patterns of different domains were implemented in a web based genome browser available at http://bar.utoronto.ca/efp_arabidopsis/cgi-bin/efpWeb.cgi?dataSource=Shoot_Apex. The source data underlying Figs. 1c-d, 2a, 3b, Supplementary Figs. 5, 7a, and Supplementary Data 1, 2, 4-6, 9-11 are provided as a Source Data file. The authors declare that all other data supporting the findings of this study are available within the manuscript and its Supplementary files or are available from the corresponding author upon request.

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

## Acknowledgements

We thank Drs. Yuval Eshed, Tengbo Huang, Thomas Laux, David Smyth, and Hirokazu Tsukaya for seeds. This work was supported by National Natural Science Foundation of China (NSFC) Grants 31430010, 31825002, 31861143021, and 31861130355, National Basic Research Program of China (973 Program) Grant 2014CB943500, and a Royal Society Newton Advanced Fellowship (award NAF/R1/180125) to Y.J., by the National Transgenic Science and Technology Program Grant 2016ZX08010-002, NSFC Grant 31770311, and Youth Innovation Promotion Association of CAS (2017139) to C.T., by NSFC Grant 31871245, University of CAS (Grant 110601M206), and the Beijing NOVA Program (Z161100004916107) to Y.W., and by the US NSF 2010 Project Grant MCB-0929349 to E.M.M. and Y.J. The Meyerowitz Laboratory is also supported by the Howard Hughes Medical Institute.

## Author contributions

Y.J. and E.M.M. conceived the research. Y.J. designed the research. Y.W., J.H., C.T., and Y.J. generated transgenic lines for translatome profiling. J.W., C.T., and Y.W. performed expression analysis of reporter lines. C.T., J.H., and Y.W. performed translatome pro-filing. C.T. performed informatic analysis. H.Y., Q.D., and N.J.P. setup the eFP browser. C.T., Y.J., and E.M.M. wrote the manuscript with assistance from all coauthors.

## Additional information

**Competing interests:** The authors declare no competing interests.

