## [Peer Review File · Nature Communications]

Reviewers' comments:

Reviewer #1 (Remarks to the Author):

The manuscript entitled “A gene expression map of shoot domains reveals new regulatory mechanisms” describes the analysis of cell-specific translomes of different shoot domains, including both shoot apical meristem (SAM) and leaf domains. They use a total of nine cell-type specific promoters to drive the expression of the ribosomal protein L18 generating different transgenic lines. Translating Ribosome Affinity Purification (TRAP) followed by RNA-seq allowed them to generate a spatial map of the covering key SAM and leaf domains. Experiments were performed using three biological replicates for each shoot or leaf domain with good reproducibility. The data generated seems to be robust and the in silico and statistical analysis of the data appears rigorous. Thus, the overall data generated in this study is novel and constitute a nice resource for plant biologist, especially for those focused on leaf development. However, there are some issues that need to be addressed in order to improve the of the manuscript.

Major points

1- The discussion is quite poor and vague. It needs to be substantially enriched by comparing the data generated in this study with previous shoot translome studies such as those conducted by Mustroph et al., 2009 (doi.org/10.1073/pnas.0906131106); Liu et al, 2013 (doi.org/10.1105/tpc.113.114769); Merchante et al, 2015 ([doi: 10.1016/j.cell.2015.09.036](https://doi.org/10.1016/j.cell.2015.09.036)), Missra et al 2015 (doi.org/10.1105/tpc.15.00546) and Hsu et al, 2016 (doi.org/10.1073/pnas.1614788113)

2- I wonder why the authors focused their analysis only in previously annotated lncRNAs. Since many lncRNAs are expressed in a cell specific manner, the data generated in this study would be a good source for the discovery of novel lncRNAs that have not been previously annotated in Arabidopsis due to dilution when using whole organs.

3- Please provide a better description for han and rbe mutants, indicating which type of mutants are they, where is the mutation and mention whether or not they are null mutants.

Reviewer #2 (Remarks to the Author):

This paper compares translating RNAs from shoot apical meristem with leaf. Overall, the novelty of the findings is not the level required to publish in Nature Communication. Any comparing data set can be useful; but whether this study uncovers advancement in our understanding of shoot apical meristem function and control is questionable. The authors have not experimentally demonstrated the function of the highlighted transcriptional factors. It would be interesting to confirm that the identified transcription factors play a role in a crop plant.

Reviewer #3 (Remarks to the Author):

A gene expression map of shoot domains reveals new regulatory mechanisms

Tian et al., 2018 – Nature Communications

This manuscript describes gene expression analysis in different domains of the Arabidopsis shoot apical meristem (SAM) and leaf primordia. Gene expression profiling using TRAPseq within each domain has been used to identify genes specifically expressed in particular cell types comprising that domain. The analysis presented in this paper is novel and represents a very useful resource for the plant development community, especially the excellent eGFP browser dataset. I feel the data are of high quality and reproducibility, and most of the major conclusions made by the authors are sound. The manuscript is quite long and contains a great deal of data, and perhaps the take-home message of the manuscript is somewhat lost amongst the plethora of information presented. However, I am in no doubt that this paper will have a good impact on the field and is of high quality. Like most papers that describe the analysis of very large datasets, only certain aspects of the results are highlighted and discussed in detail, which is to be expected. There is a slight tendency to assume a little too much specialist knowledge, both in terms of the background knowledge and in the way data are analysed and presented, so a little more careful explanation of the rationale and conclusions in the text is justified. The supplementary figure legends also need to be expanded to more fully explain the data.

I have a few issues that I would like to raise with the authors, mostly relating to interpretation and conclusions. My first point is perhaps a rather trivial one relating to the term rib meristem (RM). Generally the RM is considered to be the region of the SAM that gives rise to the pith of the stem, and in most schematic images of the SAM in publications the WUS-expression domain (organising centre; OC) is not considered to be part of the RM. A quick google image search will show that in most cases the OC is indicated as the region immediately above the RM. I would suggest that the

authors rephrase this as they have clearly analysed gene expression within the OC and not the subtending RM.

On Page 6 line 13 the authors state that UFO is more highly enriched in the WUS and CLV domains, consistent with the expanded UFO' reporter domain. This is a little confusing. The authors are making the point that UFO is expressed in the WUS and CLV3 domains, therefore validating the fact that the UFO expression pattern they observe with their reporter is larger than what has been reported previously (ie. it encompasses the WUS and CLV3 domains). This needs clarifying/ rephrasing to make this point clear.

In Figure 1B the authors present gene expression data to show that the marker gene's endogenous expression level is high in each of the domains delineated by reporter gene expression. While most genes are expressed highly within their own domain, this is not the case for UFO in the UFO' domain, and AS2 only shows a weak level of enrichment in the AS2 domain compared to other samples (as does AS1). Can the authors explain these caveats?

It is also apparent that some genes expressed in the AS1 domain (ie. the whole leaf primordium) are not detected in either the adaxial- or abaxial-specific datasets obtained using the AS2 or FIL reporters, which together encompass the whole leaf primordium like AS1. This is hard to interpret. Moreover, there are more unique genes in the AS1 domain than for AS2 and FIL. Given that AS1 encompasses the whole leaf primordium, including the AS2-associated adaxial cells and FIL-associated abaxial cells, one would expect genes expressed in the latter domains to also be expressed in the AS1 domain. This is also hard to interpret, as it would seem impossible to me for genes expressed in the AS2 or FIL domains to not also be expressed in the AS1 domain and vice-versa, as there is overlap in the cells used for transcriptome profiling. I suspect this is due to the threshold for what constitutes 'unique' expression (ie. the threshold of differential expression). Can the authors comment on this?

I am unclear as to what exactly what is being shown in FigS8. Are the heat maps in each 'box' the genes relating to the specific GO category? Is the yellow colour the degree of unique expression or the degree of enrichment across all the datasets? The authors should also explain what the LR (log2 odds ratio) in Fig 9 indicates, and what a positive and negative (red/ blue) values mean precisely.

The PCA analysis and clustering in Fig.2B,C shows very nicely how the datasets correlate/ do not correlate with one another. They take this further by correlating the datasets with publically available spatial gene expression datasets. In a way, I feel this weakens the manuscript, as the publically available datasets are not of as high resolution as those presented in this paper, and hence the correlation is a little muddy. For example, all domains including the WUS and CLV3 domains show strongest correlation with leaf tissue, despite these datasets and those for AS1/AS2/FIL showing more discrete separation in earlier analysis presented in the manuscript. There are some rather tenuous attempts to explain other weak correlations, but I do not really see the value in this. On Page 9 line 19 the authors state that boundary zone cells correlate most closely with root cells, suggesting that boundary cells 'may be less differentiated'. I do not see how this conclusion can be supported, as root cells are themselves differentiated (though obviously the nature of this differentiation is different between root and shoot cells). There is also a lack of correlation between ATML1 data and the epidermal dataset. Again, I think this is issue of the author's superior resolution compared to other datasets. I would exclude Fig 2D, as I think it detracts from the story.

The section on differential splicing and long non-coding RNAs is interesting. Fig3 A needs to show the gene models (at DNA level) for the two genes analysed (not just the splice variants) and the meaning of the grey bars needs to be explained. Fig 4B also needs better explanation in the legend or text as to exactly what is being shown.

The co-expression module analysis is very informative and I found to be one of the most interesting aspects of the work, and crucially there is experimental evidence that largely supports the authors' conclusions about the roles of some of the components in axillary and accessory bud formation. Relating to Figure 4 the text states that RBE is specifically expressed in boundaries but it is apparent from the data in Fig 4C that it is also expressed in the PZ.

The authors state on page 13 line 16 that cells within the UFO domain are actively forming organs and differentiating. I would question whether any real differentiation takes place in the PZ. Certainly, cells are specified for organ formation and enter a developmental trajectory that leads to terminal differentiation as part of leaf formation, but at this early stage the cells are not cytologically or histologically distinct from neighbouring PZ cells that have not yet been specified for organ formation, except for their altered division planes.

In general the types of genes that are enriched in each domain fit with what is expected from their known biological function. However, the analysis has also revealed some very interesting and hitherto unknown processes that may be going on in those cells, and this is an important aspect of this manuscript. Some of these are harder to understand than others, such as the enrichment of photosynthesis-associated genes in the WUS domain, which is unexpected. Some of these are very intriguing, such as the response to temperature stimulus in the WUS domain, which might represent a key regulatory module that controls stem cell proliferation (and hence overall plant growth) to environmental temperature.

The authors state on page 16 line 6 that the adaxial domain is enriched with photosynthetic functions, but do not indicate where this data is shown. Please amend.

The discussion, like most I have read in Nature Comms, is rather short. I also feel it is more of a results recap than a discussion, and I feel it could be improved. The authors focus on RBE and HAN, which are important, but these results are not key to the manuscript. Also, the functional diversification of KNOX and BELL genes is mentioned but this is not shown in the paper and the authors refer to Fig2e,f but these do not exist. Perhaps they have been removed from an earlier version of the manuscript, and perhaps can be added back in.

Overall, this manuscript is very technically detailed and contains a great deal of information, yet it is well-presented and interesting to read. I have no doubt it will be a valuable resource for those working in this field and I expect it will be well-cited.

Very Minor points:

FigS1 legend WUS>>GFR-ER should be WUS>>GFP-ER

Fig S12d legend wrongly annotated

Point-by-point Response to the Reviewers' Comments

We would like to thank the reviewers for the generally constructive comments. Based the comments from the reviewers, we have revised the manuscript as detailed below.

Reviewer #1

The manuscript entitled “A gene expression map of shoot domains reveals new regulatory mechanisms” describes the analysis of cell-specific transcriptomes of different shoot domains, including both shoot apical meristem (SAM) and leaf domains. They use a total of nine cell-type specific promoters to drive the expression of the ribosomal protein L18 generating different transgenic lines. Translating Ribosome Affinity Purification (TRAP) followed by RNA-seq allowed them to generate a spatial map of the covering key SAM and leaf domains. Experiments were performed using three biological replicates for each shoot or leaf domain with good reproducibility. The data generated seems to be robust and the in silico and statistical analysis of the data appears rigorous. Thus, the overall data generated in this study is novel and constitute a nice resource for plant biologist, especially for those focused on leaf development. However, there are some issues that need to be addressed in order to improve the of the manuscript.

1. The discussion is quite poor and vague. It needs to be substantially enriched by comparing the data generated in this study with previous shoot transcriptome studies such as those conducted by Mustrup et al., 2009 (doi.org/10.1073/pnas.0906131106); Liu et al, 2013 (doi.org/10.1105/tpc.113.114769); Merchante et al, 2015 ([doi: 10.1016/j.cell.2015.09.036](https://doi.org/10.1016/j.cell.2015.09.036)), Missra et al 2015 (doi.org/10.1105/tpc.15.00546) and Hsu et al, 2016 (doi.org/10.1073/pnas.1614788113)

Reply: Thanks for the comment. We have substantially revised the Discussion. Among other changes, we have discussed relationship with other transcriptome and Ribo-seq studies.

2. I wonder why the authors focused their analysis only in previously annotated lncRNAs. Since many lncRNAs are expressed in a cell specific manner, the data generated in this study would be a good source for the discovery of novel lncRNAs

that have not been previously annotated in Arabidopsis due to dilution when using whole organs.

Reply: Thanks for the comment. We have substantially expanded our analysis of lncRNA expression. On one hand, we have included additional annotated lncRNA described in RepTAS, Liu et al., 2012, and Song et al., 2016. On the other hand, we predicted novel lncRNAs using our expression data. We found 13 novel lncRNAs, and 21 lncRNAs that overlap with but extend previous annotations at the 5' or/and the 3' ends. We have now also included pseudogenes in our analysis, and found 125 pseudogenes in our transcriptome dataset. The numbers, figures and supplementary tables have been updated with the new results.

3- Please provide a better description for han and rbe mutants, indicating which type of mutants are they, where is the mutation and mention whether or not they are null mutants.

Reply: Thanks for the comment. We have added the descriptions of *han-2*, *han-30* and *rbe-2*. *han-2* and *han-30* are point mutations, which lead to amino acid changes in zinc finger or HAN motifs, respectively. *rbe-2* is a T-DNA insertion line in the coding sequence of *RBE*. Please see details on Page 12-13, Page 19-20 and also Supplementary Fig. 12.

Reviewer #2

This paper compares translating RNAs from shoot apical meristem with leaf. Overall, the novelty of the findings is not the level required to publish in Nature Communication. Any comparing data set can be useful; but whether this study uncovers advancement in our understanding of shoot apical meristem function and control is questionable. The authors have not experimentally demonstrated the function of the highlighted transcriptional factors. It would be interesting to confirm that the identified transcription factors play a role in a crop plant.

Reply: Thanks for the suggestion. We provided functional analysis of two transcription factor-encoding genes, *HANABA TARANU* and *RABBIT EARS*, as proof of concept to support the transcriptome analysis. Although the data are insufficient for two manuscripts, our analysis clearly showed that these two genes function in axillary bud formation. We agree that testing our findings in crops will be very useful. We indeed expect our findings to shed light on crop improvement. Nevertheless, translating findings in this work into crops is far beyond the scope of this manuscript.

Reviewer #3

This manuscript describes gene expression analysis in different domains of the Arabidopsis shoot apical meristem (SAM) and leaf primordia. Gene expression profiling using TRAPseq within each domain has been used to identify genes specifically expressed in particular cell types comprising that domain. The analysis presented in this paper is novel and represents a very useful resource for the plant development community, especially the excellent eGFP browser dataset. I feel the data are of high quality and reproducibility, and most of the major conclusions made by the authors are sound. The manuscript is quite long and contains a great deal of data, and perhaps the take-home message of the manuscript is somewhat lost amongst the plethora of information presented. However, I am in no doubt that this paper will have a good impact on the field and is of high quality. Like most papers that describe the analysis of very large datasets, only certain aspects of the results are highlighted and discussed in detail, which is to be expected. There is a slight tendency to assume a little too much specialist knowledge, both in terms of the background knowledge and in the way data are analysed and presented, so a little more careful explanation of the rationale and conclusions in the text is justified. The supplementary figure legends also need to be expanded to more fully explain the data.

Overall, this manuscript is very technically detailed and contains a great deal of information, yet it is well-presented and interesting to read. I have no doubt it will be a valuable resource for those working in this field and I expect it will be well-cited.

1. I have a few issues that I would like to raise with the authors, mostly relating to interpretation and conclusions. My first point is perhaps a rather trivial one relating to the term rib meristem (RM). Generally the RM is considered to be the region of the SAM that gives rise to the pith of the stem, and in most schematic images of the SAM in publications the WUS-expression domain (organising centre; OC) is not considered to be part of the RM. A quick google image search will show that in most cases the OC is indicated as the region immediately above the RM. I would suggest that the authors rephrase this as they have clearly analysed gene expression within the OC and not the subtending RM.

Reply: Thank you very much for the comment. We have renamed the *WUS*-expression domain as Organizing center (OC), and explained its relationship with the rib meristem.

2. On Page 6 line 13 the authors state that UFO is more highly enriched in the WUS and CLV domains, consistent with the expanded UFO' reporter domain. This is a little confusing. The authors are making the point that UFO is expressed in the WUS and CLV3 domains, therefore validating the fact that the UFO expression pattern they observe with their reporter is larger than what has been reported previously (ie. it encompasses the WUS and CLV3 domains). This needs clarifying/ rephrasing to make this point clear.

Reply: Thanks for the comment. It is likely that the *UFO* reporter we used has an enlarged expression domain than the endogenous *UFO*. To reflect this fact, we used *UFO'* to denote this enlarged domain. It was likely because of the *UFO'* enlargement, we observed endogenous *UFO* expression at higher levels in the *WUS* and *CLV3* domains than in the *UFO'* domain. There are overlaps between *UFO* and *WUS/CLV3* domains.

3. In Figure 1B the authors present gene expression data to show that the marker gene's endogenous expression level is high in each of the domains delineated by reporter gene expression. While most genes are expressed highly within their own domain, this is not the case for UFO in the UFO' domain, and AS2 only shows a weak level of enrichment in the AS2 domain compared to other samples (as does AS1). Can the authors explain these caveats?

Reply: Thanks for pointing this out. As mentioned above and in the manuscript, the *UFO'* domain is larger than the endogenous *UFO* expression domain, making *UFO* expression diluted. Based on *in situ* hybridization results (Iwakawa et al. 2007 *Plant J.*), *AS2* expression extends into older leaves but is relatively weak and diffused in P₅ and older leaves. The *AS2* reporter line we used is expressed in similar cells, but a bit more uniform between young and mature leaf adaxial domains. By contrast, the expression of *AS1* is mostly detected in young leaves (Iwakawa et al. 2007 *Plant J.*; Tian et al. 2014 *Mol. Syst. Biol.*). These two biases may explain the enrichment of strong *AS2* in the *AS1* domain. The expression of the *ATML1* reporter is epidermis-specific, also strongly overlapping with *AS2*, making *AS2* expression highly enriched in this domain. We have discussed this issue in the manuscript.

4. It is also apparent that some genes expressed in the AS1 domain (ie. the whole leaf primordium) are not detected in either the adaxial- or abaxial-specific datasets obtained using the AS2 or FIL reporters, which together encompass the whole leaf primordium like AS1. This is hard to interpret. Moreover, there are more unique genes in the AS1 domain than for AS2 and FIL. Given that AS1 encompasses the whole leaf primordium, including the AS2-associated adaxial cells and FIL-associated abaxial cells, one would expect genes expressed in the latter domains to also be expressed in the AS1 domain. This is also hard to interpret, as it would seem impossible to me for genes expressed in the AS2 or FIL domains to not also be expressed in the AS1 domain and vice-versa, as there is overlap in the cells used for transcriptome profiling. I suspect this is due to the threshold for what constitutes 'unique' expression (ie. the threshold of differential expression). Can the authors comment on this?

Reply: Thanks for the comment. Although spatially, the AS1 domain covers both the AS2 and the FIL domains, the AS1 promoter is only active in young leaf primordia earlier than P₆. In older leaves, AS1 is only expressed in the vascular region (Iwakawa et al. 2007 *Plant J.*; Tian et al. 2014 *Mol. Syst. Biol.*). This temporal difference likely contributed to the unexpectedly large number of AS1 domain-enriched genes. We have discussed this possibility in Page 40.

5. I am unclear as to what exactly what is being shown in FigS8. Are the heat maps in each 'box' the genes relating to the specific GO category? Is the yellow colour the degree of unique expression or the degree of enrichment across all the datasets? The authors should also explain what the LR (log₂ odds ratio) in Fig 9 indicates, and what a positive and negative (red/ blue) values mean precisely.

Reply: Thanks for the question. We have added a detailed legend for Supplementary Fig. 8. The colors represent z-score, i.e. the degree of relative expression. In addition, we have added a detailed description of the formula for LR calculation in the Methods section, and have added an explanation of the colors in Supplementary Fig. 9 in the legend. In brief, red color means enrichment and blue indicates depletion.

6. The PCA analysis and clustering in Fig.2B,C shows very nicely how the datasets correlate/ do not correlate with one another. They take this further by correlating the

datasets with publically available spatial gene expression datasets. In a way, I feel this weakens the manuscript, as the publically available datasets are not of as high resolution as those presented in this paper, and hence the correlation is a little muddy. For example, all domains including the WUS and CLV3 domains show strongest correlation with leaf tissue, despite these datasets and those for ASI/AS2/FIL showing more discrete separation in earlier analysis presented in the manuscript. There are some rather tenuous attempts to explain other weak correlations, but I do not really see the value in this. On Page 9 line 19 the authors state that boundary zone cells correlate most closely with root cells, suggesting that boundary cells ‘may be less differentiated’. I do not see how this conclusion can be supported, as root cells are themselves differentiated (though obviously the nature of this differentiation is different between root and shoot cells). There is also a lack of correlation between ATML1 data and the epidermal dataset. Again, I think this is issue of the author’s superior resolution compared to other datasets. I would exclude Fig 2D, as I think it detracts from the story.

Reply: Thank you for the positive feedback. We agree that the publicly available data were all derived from microarray-based studies, which was sufficiently different from the RNA-seq approach used in this study. In addition, different cell/domain-enrichment approaches were used, making such comparison not really meaningful. We have removed such comparisons

7. The section on differential splicing and long non-coding RNAs is interesting. Fig3 A needs to show the gene models (at DNA level) for the two genes analysed (not just the splice variants) and the meaning of the grey bars needs to be explained. Fig 4B also needs also needs better explanation in the legend or text as to exactly what is being shown.

Reply: Thank you for the comment. We have included gene models in Figure 3A. We have also supplemented the figure legends.

8. The co-expression module analysis is very informative and I found to be one of the most interesting aspects of the work, and crucially there is experimental evidence that largely supports the authors’ conclusions about the roles of some of the components in axillary and accessory bud formation. Relating to Figure 4 the text states that RBE

is specifically expressed in boundaries but it is apparent from the data in Fig 4C that it is also expressed in the PZ.

Reply: Thank you for the positive feedback. We have revised the description as “RBE is enriched in boundary cells” on Page 12.

9. The authors state on page 13 line 16 that cells within the UFO domain are actively forming organs and differentiating. I would question whether any real differentiation takes place in the PZ. Certainly, cells are specified for organ formation and enter a developmental trajectory that leads to terminal differentiation as part of leaf formation, but at this early stage the cells are not cytologically or histologically distinct from neighbouring PZ cells that have not yet been specified for organ formation, except for their altered division planes.

Reply: Thanks for the comment. We have revised accordingly

10. In general the types of genes that are enriched in each domain fit with what is expected from their known biological function. However, the analysis has also revealed some very interesting and hitherto unknown processes that may be going on in those cells, and this is an important aspect of this manuscript. Some of these are harder to understand than others, such as the enrichment of photosynthesis-associated genes in the WUS domain, which is unexpected. Some of these are very intriguing, such as the response to temperature stimulus in the WUS domain, which might represent a key regulatory module that controls stem cell proliferation (and hence overall plant growth) to environmental temperature.

Reply: Thanks for the comment.

11. The authors state on page 16 line 6 that the adaxial domain is enriched with photosynthetic functions, but do not indicate where this data is shown. Please amend.

Reply: Thanks for the comment. We have provided adaxial domain enriched GO terms in Supplementary Table S11.

12. The discussion, like most I have read in Nature Comms, is rather short. I also feel it is more of a results recap than a discussion, and I feel it could be improved. The authors focus on RBE and HAN, which are important, but these results are not key to the manuscript. Also, the functional diversification of KNOX and BELL genes is

mentioned but this is not shown in the paper and the authors refer to Fig2e,f but these do not exist. Perhaps they have been removed from an earlier version of the manuscript, and perhaps can be added back in.

Reply: Thanks for the comment. We have substantially revised the Discussion. Among other changes, we have deleted the part related to *KNOX* and *BELL* genes.

13. FigS1 legend WUS>>GFR-ER should be WUS>>GFP-ER

Reply: Thanks for pointing this out. We have revised the typo.

14. Fig S12d legend wrongly annotated

Reply: Thanks for the comment. We have this corrected.

REVIEWERS' COMMENTS:

Reviewer #1 (Remarks to the Author):

The manuscript entitled "A Gene Expression Map of Shoot Domains Reveals New Regulatory Mechanisms" has been significantly revised following the comments made by the reviewers to the previous version. This manuscript constitutes a significant advance in the analysis of cell specific gene expression analysis of *Arabidopsis* aerial tissues and provides high quality information about cell specific transcriptomes.

In this new version, the authors have expanded the discussion by comparing their data with previous transcriptome analysis performed in *Arabidopsis* plants. In addition, they have expanded the analysis of long non coding RNAs (lncRNAs) using de novo assembly and identified a list of novel lncRNAs. With this new analysis they found that almost half of identified lncRNAs were enriched in a specific shoot domain. They have also expanded the discussion of lncRNAs.

In this manuscript, the authors also functionally analyzed two genes, RBE and HAN, by using mutant lines. In this new version, they have provided a better description and characterization of these mutants, including the genotypic analysis as a supplemental figure.

In my opinion the manuscript has been significantly improved by including new analysis, description and discussion of the presented results. The data provided here will be highly useful for the community working in plant development and, in particular, for those investigating the mechanisms of specification of shoot domains.

Maria Eugenia Zanetti

Reviewer #3 (Remarks to the Author):

The authors have addressed the comments made in my first review and I now recommend the manuscript for publication.